# Recent Advances in Transcription Factors Biomarkers and Targeted Therapies Focusing on Epithelial–Mesenchymal Transition

**DOI:** 10.3390/cancers15133338

**Published:** 2023-06-25

**Authors:** Kai-Ting Chuang, Shyh-Shin Chiou, Shih-Hsien Hsu

**Affiliations:** 1Graduate Institute of Medicine, College of Medicine, Kaohsiung Medical University, Kaohsiung 80708, Taiwan; ti880808@gmail.com; 2School of Medicine, College of Medicine, Kaohsiung Medical University, Kaohsiung 807, Taiwan; 3Division of Pediatric Hematology and Oncology, Department of Pediatrics, Kaohsiung Medical University Hospital, Kaohsiung Medical University, Kaohsiung 807, Taiwan; 4Graduate Institute of Clinical Medicine, College of Medicine, Kaohsiung Medical University, Kaohsiung 807, Taiwan; 5Center of Applied Genomics, Kaohsiung Medical University, Kaohsiung 807, Taiwan; 6Department of Medical Research, Kaohsiung Medical University Hospital, Kaohsiung Medical University, Kaohsiung 807, Taiwan

**Keywords:** transcription factor, epithelial–mesenchymal transition, cancer targeted therapy

## Abstract

**Simple Summary:**

The study discusses the potential of targeting transcription factors (TFs) for cancer therapy and provide a systematic classification of various types of TFs involved in the epithelial–mesenchymal transition (EMT) process based on their DNA-binding domain (DBD) structure and highlights some of the main TFs that have the potential to be cancer biomarkers or targeted therapies. Various strategies for targeting TFs, such as small molecules, RNA interference, and immunotherapies, and examples of drugs currently in clinical trials are listed in this study, providing an insight into the role of TFs in EMT and targeted therapies.

**Abstract:**

Transcription factors involve many proteins in the process of transactivating or transcribing (none-) encoded DNA to initiate and regulate downstream signals, such as RNA polymerase. Their unique characteristic is that they possess specific domains that bind to specific DNA element sequences called enhancer or promoter sequences. Epithelial–mesenchymal transition (EMT) is involved in cancer progression. Many dysregulated transcription factors—such as Myc, SNAIs, Twists, and ZEBs—are key drivers of tumor metastasis through EMT regulation. This review summarizes currently available evidence related to the oncogenic role of classified transcription factors in EMT editing and epigenetic regulation, clarifying the roles of the classified conserved transcription factor family involved in the EMT and how these factors could be used as therapeutic targets in future investigations.

## 1. Introduction

Cancer comprises more than 100 diseases that progress over time and are characterized by uncontrolled cell division. According to the World Health Organization, cancer accounted for nearly 10 million deaths in 2020. The epithelial–mesenchymal transition (EMT) is important for cancer development [1], and it involves the transformation of cuboidal, non-motile epithelial cells into a loosely organized, fibroblast-like mesenchymal phenotype with reduced intercellular adhesion, loss of apical-basal polarity, and increased motility and invasiveness.

There are three types of EMTs. Type 1 occurs during embryogenesis, while type 2 occurs during wound healing and fibrosis. Type 3 occurs in cancer and represents the first step toward cancer progression to the metastatic stage; here, cancer cells acquire the ability to erode the extracellular matrix, migrate, and eventually enter the bloodstream [2]. EMT activation is involved in the malignant progression of many cancers and can induce various cancer cell features, including the acquisition of a stem cell-like phenotype, enhancement of cancer cell metastasis, resistance to chemotherapy, and antigenic escape [3,4]. Despite the vast amount of data showing the importance of EMT in cancers, its character in vivo is debated since it is difficult to perform genetic fate-mapping or lineage-tracking of cancer in human cancer tissue. In addition, through the analysis of several studies on embryonic development, wound healing, and human tumors, David Tarin raised questions about the existence of EMT and its role in carcinogenesis in adult organisms. In contrast, in the perspective of Thompson et al., they assert that multiple studies have collectively provided evidence of coordinated molecular changes between the epithelial and mesenchymal states. They also state that a comprehensive molecular analysis of individual cells in actual tumors is needed to prove the existence and importance of EMT in carcinoma progression in vivo [5]. Currently, multiple studies have emerged to delve into this issue and it is proved in studies that targeting EMT-associated factors in cancer is a promising strategy [6].

Targeting TFs that engage in the EMT process as a cancer treatment strategy has been discussed in multiple studies [7,8]. However, there have been few reviews of TFs targeted therapies based on a systematic classification of TFs. Based on the research of Wingender et al. [9], because TFs are found to be able to recognize regulatory elements in promoters and enhancers on different DBDs by their own DNA–protein recognition code (Table 1), TFs are classified based on the different DNA-binding domains (DBDs) including the basic domain, zinc-coordination DBD, helix-turn-helix domain, etc.; these domains are called a rank of Superclass in Wingender’s study. Wingender’s team classified the structure of DBD into several ranks (superclass, class, family, subfamily, genus, species) that serve as the main framework of this study, and the folding of TF and the way it establishes the DNA interacting interface characterized as a class (for example, bZIP factors, Basic helix-loop-helix factors (bHLH) factors, etc.); TF in a class are subsumed to families or subfamilies (e.g., Jun, FOS, FOX, and c-Myb) [10,11]. The goal of this review is to categorize and list the TFs involved in the EMT according to their various DBDs and to indicate their role in carcinogenesis biomarkers and targeted therapies.

## 2. The Role of EMT-TFs in Cancer

In tumor progression, classical EMT is characterized by decreased intercellular adhesion, loss of epithelial markers (such as E-cadherin and claudins), and acquisition of mesenchymal markers (such as vimentin and N-cadherin) [12]. The process is regulated by several key TFs, including Snail family proteins (including Snail1, Snail2), Zinc finger E-box binding (ZEB) homeobox family proteins, and Twist family proteins, which suppress the expression of genes linked to the epithelial state and simultaneously promote the expression of genes associated with the mesenchymal state. EMT has been proven to play a crucial role in various stages of embryonic development and result in the accumulation of extracellular matrix in fibrosis, as well as driving the progression of carcinomas towards a metastatic state [13]. Several factors cooperate to induce EMT which leads to inflammation and fibrosis in cancer. For example, TGF-β1, TNF-α, and hypoxia work together to initiate EMT by activating Snai1 through various mechanisms, with NF-κB activation playing a central role [13]. In addition, the partial activation of EMT by EMT-TFs are also reported to promote increased motility of cancer cells, whether through collective migration in cell clusters or as individual cells, and facilitate invasion and dissemination [14].

Interestingly, apart from their crucial role in the classical EMT program that regulates cancer invasion, EMT-TFs exhibit multiple characteristics in other aspects of cancer progression, including tumor initiation and chemoresistance. Firstly, EMT-TFs are associated with features that facilitate malignant progression, including evasion of senescence, DNA repair, and anti-apoptotic phenotypes [15]. Furthermore, EMT-TFs regulate the expression of pro-inflammatory and immunosuppressive cytokines in cancer cells, thereby modulating the tumor microenvironment [16]. Additionally, EMT-TFs seem to exhibit non-redundant functions that are often specific to different tissues and tumor types. For instance, the effects of Snai1 and ZEB1 on metastasis can vary depending on the type of cancer. Similarly, different EMT-TFs within the same family, such as ZEB1 and ZEB2, can have contrasting roles in tumor aggressiveness [17]. In the following section, we will classify some EMT-TFs according to their structure and introduce their characteristics in cancers.

## 3. Targeting TFs of Basic Domain

The basic domain TFs have the property of contacting DNA via a basic region with a random structure in solution and transferring to an alpha-helically folded structure upon binding to DNA. This domain is dominated by two types of proteins: bZIP (basic motif leucine zipper) and bHLH (basic motif helix-loop-helix) proteins [10].

### 3.1. bZIP Factors

The proteins identified as bZIP factors are characterized by a leucine zipper region that mediates dimerization with another bZIP domain and a basic region that binds to DNA. The basic region contains positively charged amino acids that interact with the negatively charged phosphate backbone of DNA, allowing for DNA-binding specificity. In contrast, the leucine zipper region contains leucine residues that form an alpha-helix and interact with another alpha-helix from a dimerization partner to form a coiled-coil structure [11]. Jun, FOS, and BACH1 are categorized in this family.

#### 3.1.1. Jun and FOS

The Jun gene and Fos gene encoded the Jun and Fos TF individually. The Jun subfamily is categorized into several genera, including c-Jun, JunB, and JunD; while c-Fos, FosB, Fra-1, and Fra-2 are classified under the Fos subfamily. C-Jun and c-Fos proteins are subunits of the AP-1 (Activator Protein-1) complex, which modulates a variety of cellular signaling pathways and controls important cellular processes such as differentiation, proliferation, migration, and apoptosis [18,19]. A large body of evidence supports the pivotal role of AP-1 in various primary cancer tissues, including lung cancer, colorectal adenocarcinoma, breast cancer (BC), and Hodgkin lymphoma [20,21,22,23].

Several studies have examined the roles of Jun and Fos in EMT. C-Jun forms an axis with SNAI2 in lung adenocarcinoma, acting as an essential regulator of EMT [24]. Furthermore, in the radio-resistant human nasopharyngeal carcinoma cell line CNE-2R, the inhibition of c-Jun expression reduces EMT, cell migration, and invasion [25]. Fra-1, which is encoded by Fosl1, induces EMT-related gene expression alternations in mammary epithelial cells. These alterations promote the cellular acquisition of mesenchymal, invasive, and tumorigenic properties [26]. The alteration of the EMT process by Fosl1 promotes prostate cancer development. Moreover, according to Feldker et al., Jun and Fosl1 interact with the TF ZEB1 and create a transactivation complex that primarily activates tumor-promoting genes, thereby enhancing its role as a suppressor of epithelial genes [27,28].

Strategies of selective c-Jun/c-FOS antagonism targeting different sites have been discussed. For example, DNA 12-O-tetradecanoylphorbol-13-acetate (TPA) response element (TRE) sites (MLN944 [29,30], SR11302 [31], and Veratramine [32]), the c-Jun DBD (T-5224 [33,34]), the Lucine zipper interface, and the entire bZIP domain. Despite prominent research demonstrating the potential of inhibiting the expression of c-Jun and c-FOS or blocking their related binding domain as therapeutic targets for cancers, no AP-1 family member inhibitor has been approved for practical clinical use [19].

#### 3.1.2. BACH1

BACH1 belongs to the NFE2 subfamily and combines broad-complex, tramtrack, bric-a-brac, and bZip domains. Recent studies have found that BACH1 promotes the advancement of several cancers, including clear-cell renal cell carcinoma [35], esophageal squamous cell carcinoma [36], hepatocellular carcinoma (HCC) [37], and pancreatic cancer [38].

BACH1′s role in cancer progression is vital and includes regulation of iron homeostasis and related reactions [39], regulation of the oxidative stress response, enhancement of cancer cell metastasis, and promotion of aerobic glycolysis and EMT [40]. More specifically, BACH1 suppresses the FOXA1 gene, which activates epithelial genes, and simultaneously activates SNAI2, which represses epithelial genes. This creates a feed-forward loop that promotes EMT. BACH1 overexpression significantly enhanced CDH2 (N-cadherin) promoter activity, stimulating EMT [36,41]. BACH1 is a novel therapeutic target for regulating cancer progression.

### 3.2. Basic Helix-Loop-Helix Factors (bHLH) Factors

A DNA-binding basic region is followed by a motif of two potential amphipathic alpha-helices connected by a loop, possibly an omega loop, in bHLH. The DNA-binding motif of the basic helix-loop-helix (bHLH) family is a preserved region of approximately 60 amino acids that includes two alpha-helices (the “basic” region and the “helix” region) separated by a loop. The positively charged amino acids in the bHLH domain’s basic region bind to the negatively charged phosphate groups of DNA, while the helix region helps stabilize the interaction between the bHLH domain and DNA. The bHLH domain can also form homo- or heterodimers with other bHLH proteins, providing greater specificity in DNA-binding and regulation of gene expression [10,11]. Based on phylogenetic analyses, this domain contains several families, including E2A, Twist, and HIF1α; their roles in EMT and cancer formation are discussed further below.

#### 3.2.1. E2A

The expression of E2A basic helix-loop-helix (bHLH) TFs is involved in EMT and is reportedly coexpressed with SNAIL1 in human basal-like breast tumors [42]. E2A is linked to the development of several types of cancer, including BC, colorectal cancer, ovarian cancer, cervical carcinoma, and acute lymphoblastic leukemia [42,43,44,45,46,47]. Slattery et al. found that E2A overexpression changes cell morphology, cytoskeletal arrangement, and the expression of E-cadherin and α-SMA, hallmarks of the EMT process [48].

According to López-Menéndez et al.’s recent study, E2A plays an important role in BC; it is involved in stemness, metastasis, and therapeutic resistance of BC [42]. Furthermore, in 5% of children with acute lymphoblastic leukemia (ALL), the t (1,19) chromosomal translocation specifically targets the E2A gene, resulting in an oncogenic E2A-PBX1 fusion protein and becoming a coactivator for RUNX1, resulting in unfavorable B-cell development [49]. E2A is a promising predictor of cancer clinical outcome; however, few clinical trials currently target E2A, and knowledge of E2A’s detailed functional contributions to tumor progression is still limited.

#### 3.2.2. Twist

The key Twist family members Twist-1 and Twist-2 are rarely present in healthy adult tissues but are commonly overexpressed in various human cancer tissues, including prostate, uterus, breast, liver, and skin. Twist-1 and -2 act as molecular switches that activate or suppress target genes through direct or indirect mechanisms. For instance, Twist’s C-terminal contains a “Twist box” relevant to its anti-osteogenic function. The primary difference between the two isoforms is that Twist-1 has a glycine-rich region in its N-terminal sequence, which is absent in Twist-2 [50]. Twist proteins play a critical role in the EMT; both Twist-1 and Twist-2 induce the EMT and regulate the expression of EMT-associated genes or downstream effectors, such as N-cadherin or epithelial membrane protein 3 (EMP3) [51]. Therefore, the Twist TFs regulate various target genes associated with EMT, including cell migration, self-renewal of cancer stem cells, multiple drug resistance, cell apoptosis, and immune surveillance. High Twist-1 expression enhances tumor invasion in invasive squamous cell carcinomas (SCCs); its overexpression is related to enzastaurin resistance in colon cancer cells [50]. In contrast, Twist-2 stimulates the migration and invasion of gastric cancer (GC) cells [52].

Twist is a potential therapeutic target for inhibiting cancer progression because of its rare expression in normal adult tissues. Further, Twist can be successfully inactivated by long non-coding RNAs (lncRNAs) or chemotherapeutic approaches [53]. Therefore, drugs that target Twist specifically could be used as a novel cancer treatment.

## 4. Targeting TFs of Zinc-Coordination DNA-Binding Domain

A zinc ion coordinated by two or more conserved cysteine or histidine residues in the protein structure defines zinc-coordinating domains. When a protein loses its zinc ion, it loses its ability to bind DNA. Among the TFs of this superclass, the first and most prominent members are the zinc finger proteins of polymerase III transcription factor (TFIIIA)/Krüppel type, as well as those of nuclear receptors and RNF36 [10,11].

### 4.1. Nuclear Receptors with C4 Zinc Fingers

Nuclear receptors with C4 zinc fingers represent a zinc finger motif of nuclear receptor type. Each nuclear receptor molecule contains two DNA-binding motifs that differ in size, composition, and function. Four cysteine residues coordinate one zinc ion to form a zinc finger, which binds to DNA by recognizing specific DNA sequences known as hormone response elements (HREs). The first zinc finger binds to DNA via the major groove, and the second zinc finger mediates dimerization upon DNA binding in an alpha-helix conformation. Androgen receptor (AR), retinoic acid receptor (RAR), and Snail-like TF are discussed below.

#### 4.1.1. Steroid Hormone Receptor (SHR)

SHRs can recognize specific cis-acting DNA sequences known as HRE on target genes and act as TF in humans by binding to a steroid hormone ligand, converting the hormonal stimulus into a transcription response. They play a role in cellular growth and proliferation, leading to cancer development [54,55,56]. The SHR family comprises several members including androgen receptors (AR), estrogen receptors (ER), glucocorticoid receptors (GR), and progesterone receptors (PR) [57].

ER plays a vital role in various cancer development, including breast cancer, ovarian cancer, prostate cancer, and bone cancer [58]. ER alpha (ERα) and ER beta (ERβ) have been identified as the predominant estrogen receptors. Among them, ERα has been recognized to be expressed in the majority of breast cancers and functions in suppressing EMT by inhibiting signaling transduction cascades such as TGFβ and NF-κB. This inhibition ultimately hinders the EMT process. ERα inhibits the TGFβ signaling pathway by binding to Smad2 and Smad3 which repress the proliferation of cancer cell; in addition, MTA3 (Metastatic Tumor Antigen 3), a suppressor of Snai1, is upregulated by ERα. The knockdown of ERα result in the activation of EMT. The interaction between ERα and ERβ is interesting. In ERα-positive cancers, the presence of ERβ may, in certain instances, promote EMT by interfering with ERα activity. However, this interference can also increase the susceptibility of ERα activity to inhibition, potentially enhancing the effectiveness of hormone-blocking agents [59].

PR is also reported to contribute to the regulation of EMT of mammary alveologenesis by inducing several components of cascades involved in EMT such as Wnt4 from the Wnt/β-catenin cascade, RANKL, a ligand of the NF-κB cascade. In vivo in rat mammary tumors and in vitro in human cell lines, the suppression of PR-B (B isoform of PR) result in the suppression of E-cadherin and induce the EMT [59]. Currently, selective progesterone receptor modulators (SPRMs) such as mifepristone, ulipristal acetate, Vilaprisan, and Telapristone Acetate, which are designed to competitively bind to the PR target site in a tissue-specific manner, have been developed; although none of them have shown success in endometrial cancer treatment, several SPRMs appear to be effective in treating recurrent breast cancer, and clinical trials are currently underway to investigate their potential [60,61].

GR plays a significant role as a sensor for various stress conditions, including life stress and inducible cellular stress (such as hypoxia, reactive oxygen species (ROS), and nutrient starvation as well as microenvironmental stress (such as cytokines). These stressors predominantly act through p38 MAPK-dependent phosphorylation of Ser134. In triple negative breast cancer cell model, the expression of GR and/or GR-associated target genes has been linked to pro-survival signaling, EMT, cellular migration/invasion in vitro, and metastasis in vivo [62]. The role of GR and its association with ER in breast cancer has been studied. The expression of GR is associated with poorer prognosis and promoted breast tumor cell invasion and lung metastasis in vivo [63]. Dysfunctional GR signaling in ER-negative carcinomas, particularly in tumors with BRCA gene mutations or impaired BRCA function, leads to Twist upregulation and promotion of EMT [59,64].

AR is involved in the EMT processes of prostate, breast, and bladder cancers and is associated with tumor metastasis and advanced cancer stages [65,66,67]. AR is reported to directly inhibits the E-cadherin promoter in breast cancer cell lines using an artificial transfection system and promotes metastatic dissemination in mice in vivo. The activation of AR increases the expression of markers of EMT, such as vimentin and N-cadherin, which is mediated through the upregulation of Wnt/β-catenin signaling [68]. Enzalutamide, an anti-androgen, targets AR’s ligand-binding domain to improve survival in patients with prostate cancer; thus, AR overexpression is associated with a worse prognosis [69,70]. Various novel therapeutic agents, such as the antidiabetic drug metformin or the selective estrogen receptor modulator ormeloxifene, have been tested in preclinical studies as potential EMT inhibitors for patients with prostate cancer [71].

#### 4.1.2. Retinoic Acid Receptor (RAR)

The RAR TF is a thyroid hormone receptor-related factor subfamily involved in the RA signaling pathway and regulates processes such as early neural differentiation, mesoderm development, and eye and forebrain development [72]. All-trans retinoic acid (ATRA) controls cell development by regulating gene expression through the activation of the three major isoforms of closely related RARs: RARα, RARβ, and RARγ. They bind as a heterodimer with the retinoid X receptor to the cis-acting response elements of ATRA target genes, enabling gene transcription when ATRA binds to RARs [73]. ATRA signaling pathway interruption may be responsible for the development of various hematological and non-hematological cancers, such as leukemias, skin cancer, head and neck cancer, lung cancer, BC, ovarian cancer, prostate cancer, renal cell carcinoma, pancreatic cancer, liver cancer, glioblastoma, and neuroblastoma [74]. RARα is an RAR subfamily member that downregulates the EMT process. RARα signaling inhibits EMT inside retinal pigment epithelial cells, potentially reducing fibrosis in proliferative retinal diseases [75]. Gong et al. found that RARα reduced the renal tubular cells’ EMT activity, preventing the hypoxia associated with the TGF-β/MMP-9 pathway [76]. Moreover, Liu’s study suggested that the deletion of RARβ had a protective effect against mammary gland tumorigenesis induced by Wnt1. This led to Wnt signaling inhibition in both the epithelial and stromal compartments and suppression of EMT [77]. In contrast, RARγ overexpression activates cancer stem cells’ abnormal behaviors and is associated with multidrug chemoresistance and tumor cell dissemination in colorectal cancer via activation of the Wnt/β-catenin pathway [78].

The role of RARs in EMT is controversial. According to Doi et al., RARα overexpression in mammary epithelial cells dramatically increased the mRNA levels of well-known EMT-inducing factors such as SLUG, FOXC2, ZEB1, and ZEB2 [79]. Chemokine CCL28 reportedly inhibits bone invasion and EMT in oral squamous carcinoma by inducing RARβ expression. Kimura et al. found that RARγ agonists suppressed the expression of severe EMT-associated proteins such as Smad2 and AKT, interrupting the EMT process [80]. Therefore, more studies are needed to clarify RAR’s role and how it regulates the EMT process.

### 4.2. C2H2 Zinc Finger Factors

C2H2 zinc finger factors feature a TFIIIA/Krueppel zinc finger motif, consisting of two cysteines and two histidine residues coordinating a zinc ion. In some cases, histidine is replaced with another cysteine. This zinc ion is crucial for DNA binding. Typically, the first half of the finger sequence is arranged in two antiparallel beta strands, while the second half is organized as an alpha-helix and partially as a 310-helix. The conserved phenylalanine and leucine residues create hydrophobic contacts between the beta strands and the alpha-helix, which binds to DNA via the major groove [9,10,11].

#### Snail-like

The Snail-like TF subfamily consisted of several members, including Snai1 (Snail), Snai2 (Slug), and Snai3 (Smuc). Snai1 is encoded by the SNAI1 gene, and its expression can be induced by various signaling proteins such as epidermal growth factor, bone morphogenetic proteins (BMPs), fibroblast growth factor, hepatocyte growth factor, transforming growth factor-β (TGF-β), Notch, Wnt, TNF-α, and cytokines [81], resulting in cancer metastasis and progression. Inhibiting SNAI1 expression using knockdown techniques impedes tumor growth and metastasis by augmenting the number of tumor-infiltrating lymphocytes and systemic immune responses [82]. Snai2, known as Slug, is involved in tumor metastasis, stem/progenitor cell biology, cellular differentiation, and DNA damage repair. Snai2 and its associated EMT protein reportedly enhance the Wnt signaling pathway [83]. Snai2 expression indicates a poorer prognosis and increased risk of metastasis for ovarian, breast, and lung cancers [84,85]. The Snail-like family plays a crucial role in EMT, regulating various cellular processes such as cellular differentiation, cell movements, and overall survival for various cancers.

Snail engages in interactions with diverse signaling molecules. One such interaction is observed between Snail and the Sin3A-HDAC1/2 complex through the SNAG domain. This interaction facilitates the deacetylation of histone H3 and H4, leading to the repression of the CDH1 promoter, which encodes E-cadherin [81]. HDAC1 and HDAC2 have been reported to repress the EMT process, while HDAC3 is found to interact with hypoxia-induced WDR5 and acts as a corepressor in suppressing the expression of epithelial genes. The role of HDAC inhibitors (HDACi) in EMT is still debated, with conflicting reports on whether they promote or inhibit EMT. According to Tang et al., in ovarian cancer, the identified HDACi (Vorinostat, Mocetinostat) have been reported to reverse EMT and promote epithelial differentiation by restoring the expression of E-cadherin and ErbB3. Additionally, the HDACi have demonstrated functional relevance in reversing resistance to anoikis (apoptosis caused by cell detachment) and eliminating the formation of spheroid. On the other hand, one of the HDACi, Trichostatin A, has been found to induce EMT through upregulating of SNAI1 and SNAI2 expression. According to Chałaśkiewicz et al., Trichostatin A induced the expression of SNAI1 and SNAI2 and downregulated the expression of SLC2A5, which is a key factor that encodes GLUT5 and play a role in diabetes and cancer. In addition, Trichostatin A is reported to sensitize colon cancer cells to cisplatin and oxaliplatin. Furthermore, according to Mrkvicova et al., the HDACi sodium butyrate (NaBu) upregulated E-cadherin, and sensitized both cisplatin-sensitive and cisplatin-resistant cells to cisplatin [86,87,88].

Although targeting the Snail family TF is an attractive choice as cancer treatment, some researchers consider Snail family members to be “undruggable” given the lack of effective pharmacological inhibitors. Recently, Li’s team discovered a small-molecule compound, CYD19, that targeted Snail and successfully disrupted CREB-binding protein (CBP)/p300-mediated Snail acetylation by binding to Snail and then promoting its degradation through the ubiquitin–proteasome pathway [89]. Additionally, thiolutin (THL), the non-ATPase regulatory subunit 14 (SMD14) inhibitor, reportedly suppresses the PSMD14/SNAIL axis, thereby decreasing the EMT process [90]. Further, the autophagy-derived acetyl-CoA promotes the acetylation of Snail, and using calcium/calmodulin-dependent protein kinase kinase 2 or ATP citrate lyase inhibitors might interrupt the autophagy/acetyl-CoA/acetyl-Snail axis and inhibit lung cancer metastasis [91]. These studies indicated that targeting the Snail family or its EMT-associated proteins could be a novel and promising strategy for treating cancers.

## 5. Targeting TFs of Helix-Turn-Helix Domain

The helix-turn-helix (HTH) superclass is the second-largest TF family, accounting for 27% of all human TF genes (Table 2). Most of the TFs in this classification perform essential functions in eukaryotes, such as developmental regulation and differentiation process determination. The HTH domain comprises two alpha-helices connected by a short beta turn to form a “V”. The first helix is often called the recognition helix because it makes specific contact with the DNA. The second helix helps stabilize the domain’s structure and interacts with other proteins in some cases [9,10,11].

### 5.1. Homeo Domain Factors

Approximately half of all TFs in the HTH superclass belong to the homeodomain class. The “homeobox” was a DNA sequence motif that encoded homeodomains of usually 60 amino acids in proteins that regulate development. The Homeo Domain factor is composed of three alpha-helices in a row, with the third helix primarily interacting with the major groove of the DNA and some interactions with the minor groove. The domain’s helices 2 and 3 resemble the structure of the HTH motif found in prokaryotic regulators. The homeodomain binds to DNA as a monomer, recognizing short DNA sequences, typically 5–8 base pairs in length, and frequently functions in transcriptional regulation [10].

#### 5.1.1. ZEB

The ZEB homeobox family proteins are key TFs that mediate the EMT process, similar to the roles of the Snail family and Twist family in EMT. The structure of ZEB proteins includes a homeodomain (HD) located in the middle, as well as other protein binding domains such as the SMAD interaction domain, which regulates transcription mediated by the transforming growth factor beta (TGFβ) and BMP signaling, the zinc finger domain, the coactivator binding domain, the CtBP interaction domain, and the p300-CBP-associated factor (PCAF) binding domain. These domains trigger EMT, leading to tumor progression and metastasis and inducing therapy resistance [92].

In terms of ZEB family members ZEB1 and ZEB2, ZEB1 expression is positively correlated with cytoplasmic and nuclear N-cadherin expression, whereas ZEB2 expression is positively correlated with cell membrane N-cadherin expression [93,94]. ZEB1 and ZEB2 are highly expressed in several cancers, including BC, pancreatic cancer, HCC, and lung cancer. Several factors mediate ZEBs; for example, the activation of MEK1/2, ERK1/2, Fos-related antigen 1 (Fra-1), and TGF-β enhance the expression of both ZEB1 and ZEB2, increasing tumor invasion. In addition, β-catenin translocates into the nucleus and activates the expression of ZEB1, whereas Wnt signaling and E2F1 upregulate ZEB2 expression, resulting in EMT activation and cancer progression [92,95]. Ashrafizadeh et al. considered microRNAs (miRNAs) modulators of ZEBs. The miR/ZEB1 axis can be regulated by long non-coding RNAs (lncRNAs) or circular RNAs (circRNAs), which regulate tumor malignancy in several cancers, including lung, gastric, and ovarian [96]. Huang et al. found that ZEB expression in hepatic stellate cells is reduced by nuclear receptor 4a1 (NR4A1), which inhibits the TGF-β-Smad2/3/4-ZEB signaling pathway, thus inhibiting the EMT-induced liver fibrosis [97].

#### 5.1.2. Intestine-Specific Homeobox (ISX)

The intestine-specific homeobox (ISX) is a subfamily of the paired-related HD family. It is a newly discovered proto-oncogene and is an intestine-specific transcription factor that regulates tumor progression and has been linked to a poor prognosis in patients with hepatocellular carcinoma (HCC), non-small cell lung carcinoma (NCSLC), and GC [98,99,100,101]. According to Wang et al., ISX, PCAF, and BRD4 form a complex that mediates EMT signaling by regulating Twist1 and Snail1 and promotes tumor initiation and metastasis. ISX organizes the feed-forward immune suppression mechanism involving kynurenine-AHR signaling and PD-L1 in HCC and provides the function of immune escape by HCC [102]. In addition, by directly binding to the E2 site of its promoter, ISX activated E2F transcription factor 1 (E2F1) and increased oncogenic activity in HCC [103]. Though several papers have reported that ISX is important in tumor progression HCC, the specific mechanism by which ISX is involved in carcinogenesis has yet to be fully clarified. The current findings suggest that ISX may be a promising biomarker for predicting the prognosis of HCC, but more clinical trials and randomized studies are required for practical clinical application.

### 5.2. Fork Head/Winged Helix Factors

The Fork head/winged helix factors were found by comparing the similarities between HNF-3A and fkh. The DNA-binding motif of the Fork head/winged helix factors is a winged HTH domain, also called the Forkhead domain. The DNA-binding domain is approximately 110 amino acids in length. According to the crystal structure analysis, the domain has three closely packed alpha-helices, with the third alpha-helix exposed toward the major groove of the DNA. The domain also makes minor groove contacts. When it binds to DNA, it causes 13 degrees to bend.

#### FOX

As the name implies, Forkhead-box (FOX) family proteins share a conserved common structurally related DBD, the Forkhead domain. It can regulate transcription and DNA repair, and it is involved in many stages of cell growth, including differentiation, embryogenesis, and longevity [104,105]. Furthermore, FOX TFs can be directly involved in DNA replication and discern the global replication timing program in a transcription-independent mechanism. Overall, 50 FOX members have been identified in the human genome and are classified into 19 subfamilies (FOXA to FOXS) based on the similarity of sequence [106]. The mutation of different subtypes FOX gene results in the development of different cancers, such as colorectal cancer [107], HCC, B-cell lymphomas, NSCLC, and cervical cancer. FOX TFs are crucial in the Wnt pathway, which is important in the molecular pathway promoting EMT. For example, FOXG1 significantly enhances the EMT process in HCC cells by inducing the nuclear transport of β-catenin and facilitating its retention in the nucleus [108]. Other FOX TF family members, including FOXC1, FOXC2, FOXK1, FOXQ1, and FOXM1, are associated with tumor metastasis and poorer outcomes through regulating TGFβ-induced EMT processes in various types of cancers, including mammary carcinoma, esophageal cancer, nasopharyngeal cancer, BC, cervical cancer, NSCLC, and HCC [4].

The concept of the regulation of FOX genes by microRNA (for example, miR-342, miR-204, and miR-1269) as a therapeutic target for cancer has recently gained more traction, indicating the existence of an additional level of complexity in the regulation of the FOX protein pathway. Directly targeting the FOX protein is another potential strategy for treating cancer. The silencing of FOXM1 by RNA interference, a process of sequence-specific posttranscriptional gene silencing initiated by double-stranded RNA, inhibits cell proliferation of BC and also helps overcome the resistance of tamoxifen [109]. Proteasome Inhibitors and other agents such as bioactive natural products (genistein), peptide inhibitors, or thiazole antibiotics may be prospective therapeutics that target the transcriptional activity or gene expression of FOX proteins [110].

## 6. Targeting TFs of Other All-Alpha-Helical DNA-Binding Domains

The superclass consists of TFs with DBD and alpha-helically structured interfaces that interact with DNA. Currently, only two classes have been classified in this superclass: HMG (high-mobility group) domain factors and heteromeric CCAAT-binding factors; however, only HMG proteins are structurally well-defined [9,10,11].

### 6.1. High-Mobility Group (HMG) Domain Factors

The proteins with an HMG domain had an identical structure, the HMG box, about 75 amino acids long. This domain has a typical L-shaped conformation with three alpha-helices and a long N-terminal extension of the first helix. Helix 1 and the N-terminal region create the long arm of the L, while helices 1 and 2 create the short arm. Binding to the minor groove of DNA causes significant bending of the DNA by more than 90 degrees away from the protein. The overall configuration of the DNA–protein complex resembles that of the TBP-TATA box complex [9,10,11].

#### SOX

The sex-determining region Y (SRY)-related HMG box (SOX) proteins are TFs that have been linked to the regulation of specific biological processes such as tumorigenesis, changes in the tumor microenvironment, and metastasis [111]. SOXs specifically bind and bend DNA with other TFs, modifying transcriptional activation early in transcription. In addition to regulating transcription initiation, some SOXs also regulate co-transcriptional RNA splicing. The SOX family contains over 20 members (subdivided into SOX A to SOX F), all of which play important roles in cell differentiation and tumor development. For example, SOX F (SOX7, SOX17, and SOX18) is involved in angio- and lymphangiogenesis and has been reported to be upregulated in breast and lung cancer, associated with poor outcomes [112]. SOXs are abnormally activated in nearly all types of human cancers, including BC, prostate cancer, liver cancer, renal cell carcinoma, thyroid cancer, cervical cancer, brain tumor, gastrointestinal cancer, and lung cancer [111,112,113,114,115]. The SOX family regulates the EMT process in cancer and enhances cancer cell proliferation. For instance, SOX4 expression was upregulated in TGFβ-treated cells undergoing EMT; it additionally regulates target genes associated with mesenchymal features, including N-cadherin, ADAM10, TMEM2, TNC, FZD5, neuropilin-1, and semaphorin-3A, to promote cancer cell migration, invasion, and metastasis by orchestrating the EMT process [116]. Additionally, SOX2 regulates EMT during cranial neural crest cell development and impacts the fate of cells involved in head growth during neural crest development [117]. Furthermore, SOX9, SOX10, and SOX11 are associated with increased tumor metastasis and worse overall survival by regulating the ability of cancer cells to undergo EMT and acquire mesenchymal characteristics [118].

Currently, SOX2 is the most promising target in the SOX family, with numerous clinical trials targeting SOX2 currently underway (e.g., SAHA, SOX2-derived peptide, and ZF-552SKD) [119,120,121]. On the contrary, SOX9, SOX10, and SOX11 are upregulated in basal-like BC, indicating that therapeutic strategies targeting these factors may be beneficial for this subtype of cancer, which currently lacks targeted therapy. The SOX TF family can be used as a prognostic marker for cancer, and targeted therapies are currently being developed.

## 7. Targeting TFs of Immunoglobulin Fold

The DNA-binding domains in this superclass have an immunoglobulin-like structure, with a beta-core, and a beta-sandwich architecture. The DNA-contact interface is primarily made up of loops, but it may also contain other elements of secondary structure, with DNA-binding residues extending from this interface [9,10,11].

### 7.1. Rel Homology Region (RHR) Factors

The DBD of Rel-type proteins consists of two subdomains, each comprising a beta-barrel with five loops that form a large interface with the DNA major groove. The N-terminal subdomain contains a highly conserved recognition loop that interacts with the DNA recognition element, as well as other loops. The application of loops to make main DNA contacts is suggested to provide flexibility in binding to different sequences. Two alpha-helices within the N-terminal part form strong contacts with the A/T-rich center of the B-element in the minor groove, providing additional interactions. The C-terminal domain of the RHR factor is primarily responsible for protein dimerization [10,11].

#### NF-κB–Related

The nuclear factor κB (NF-κB) is a transcriptional factor family of five subunits, including Rel (cRel), p65 (RelA, NFκB3), RelB, p105/p50 (NFκB1), and p100/p52 (NFκB2). It is essential for infection response in both immune and non-immune cells; it also regulates cell survival, differentiation, and proliferation by controlling the expression of biologically important genes such as regulators of apoptosis, stress-response genes, cytokines, etc., chemokines, growth factors, and their receptors [123]. The role of NF-κB in carcinogenesis has been discussed in several papers, and the NF-κB signaling pathway has been identified as an important pathway in pathogenesis and cancer treatment [124]. NF-κB pathway is vital for the induction of EMT in glioblastoma, BC, and nasopharyngeal carcinoma, contributing to tumor progression as well as treatment resistance [125,126,127]. The Aldo-Keto Reductase Family 1 Member B10 (AKR1B10), a unique tumor biomarker that is overexpressed in BC, stimulates the NF-κB pathway by inducing PI3K/AKT signaling and enhances the expression levels of ZEB1, Slug, and Twist, promoting EMT induction and increasing breast tumor cell dissemination.

Two NF-κB signaling pathways (canonical and non-canonical) mediate the inflammatory response and protein synthesis. Selective inhibition of canonical NF-κB could be a promising strategy in clinical therapy, given the tumor-promoting role of the pathway [128]. Taxanes have been shown to inhibit the activity of NF-κB1, thereby limiting tumor metastasis. Other drugs, including indomethacin, dexamethasone, sulindac, and tamoxifen, have been reported to inhibit the expression of NF-κB1 [167]. Bortezomib, a drug that targets NF-κB, is now in clinical trials. It is primarily an inhibitor of the 26S proteasome, a critical protease involved in the canonical NF-κB pathway. Bortezomib inhibits the activity of the Sp1 gene and disrupts the interaction of the Sp1/RelA complex, ultimately inhibiting the NF-kB1 pathway. Bortezomib inhibits tumor proliferation in various types of cancers, including gastric, breast, ovarian, and pancreatic tumors; additionally, the drug blocks the activity of NF-κB1 and reduces resistance to doxorubicin, making it a potential drug for treating patients with anthracycline-containing regimens resistant tumors [129]. Aside from bortezomib, numerous other small-molecule agents in clinical trials target the inhibition of cellular receptor adaptor protein in the NF-kB pathway. For example, Acalabrutinib (Phase 2 trial for metastatic pancreatic cancer, ovarian cancer, non-small-cell lung cancer (NSCLC)), Ibrutinib (Phase 3 trial for Metastatic Pancreatic Adenocarcinoma), Dasatinib (phase 2 trial for NSCLC, Cholangiocarcinoma, and BC), and LCL-161 (phase 2 trial for BC, ovarian cancer, small cell lung cancer) [130]. According to recent research, the expression of NF-κB1 may be partially responsible for BC chemoresistance and progression, and the activation of NF-κB1 may result in resistance to platinum preparations in ovarian cancer. Furthermore, increased expression of NF-kB1 is linked to poor survival rates and the development of chemo/radiation resistance in many malignant neoplasms, including rectal cancer, kidney cancer, GC, etc. The results showed that the NF-κB1 gene is also a predictive and prognostic marker in the treatment of cancers [129].

### 7.2. Signal Transducer and Activator of Transcription (STAT) Domain Factors

STAT proteins bind to DNA as dimers, with the DNA-binding interface formed by an eight-stranded beta-barrel, a four-helix bundle, and an alpha-helical connector region. The amino acid residues that interact with the major groove of the DNA are mostly exposed by loops that connect the beta strands of the beta-barrel and the one that connects the beta-barrel and the first helix of the “connector” region. The STAT dimer almost completely encircles the DNA double helix, much like a “pair of pliers,” with the DNA-binding interface serving as the jaws and the four-helix bundles serving as handles. The DNA undergoes a moderate bending of about 40 degrees upon binding by STAT [9,10,11].

#### STAT

The STAT signaling pathway involves various cellular processes, including cell differentiation, proliferation, and inflammation. STAT is involved in regulating the EMT pathway, especially the JAK/STAT pathway, and has been linked widely to advanced stage or treatment resistance in various types of cancer [131]. There are mainly seven STAT family members, with STAT1, STAT3, and STAT5 playing the most important roles in carcinogenesis [168]. STAT1 tends to respond to interferons (IFNs), regulating gene expression involved in multiple anticancer processes such as growth arrest, apoptosis, and immune surveillance; however, STAT1 has also been reported as an oncogene in serous papillary endometrial cancer, so the role of tumor suppressor of STAT1 is controversial and may be tumor-specific. STAT3 is a primary oncogenic transcription factor that primarily responds to interleukins (ILs), specifically IL-6, IL-10, IL-23, IL-21, and IL-11, as well as leukemia inhibitory factor and oncostatin M (OSM), regulating pro-tumorigenic processes. STAT5 is primarily activated by IL-2, granulocyte-macrophage CSF (GM-CSF), IL-15, IL-7, IL-3, IL-5, and prolactin (PRL) and regulates the expression of pro-growth and pro-survival genes. It has also been linked to breast, head and neck, prostate, and uterine cancers [131].

The JAK/STAT signaling pathway has several steps. Firstly, STAT is phosphorylated by Janus kinases (JAK), a non-receptor tyrosine-protein kinase that is a key component in the signaling pathway. Secondly, the STAT protein dimerizes and transports into the nucleus to regulate related gene expression. The pathway is known as the JAK/STAT signaling pathway [169]. The IL-6/JAK2/STAT3 pathway activates and promotes metastasis by increasing the expression of EMT-inducing TFs such as ZEB1, Snail, JUNB, and Twist-1. This pathway enhances cell motility by activating focal adhesion kinase (FAK) [132]. According to Stevens et al., chemotherapy in combination with JAK2/STAT3 inhibition synergistically affected resistant derivatives, suppressing resistance mechanisms associated with EMT in BC [133].

Inhibiting the JAK/STAT signaling pathway could be a promising strategy for cancer treatment. In addition to numerous studies on the clinical application of inhibiting JAK, several factors that negatively regulate JAK/STAT pathway signaling have been identified, including the Suppressor of Cytokine Signaling Proteins, the Protein Inhibitor of Activated STAT (PIAS), and protein tyrosine phosphatase (PTP) [134,135,136]. Currently, the direct inhibitors of STAT undergoing clinical trials target STAT3: OPB-31121, OPB-111077, and OPB-51602. These inhibitors have shown antitumor activity in hepatocellular carcinoma and leukemia and were the focus of several successful phase I trials [137]. Napabucasin (BBI-608), a novel STAT3-targeted agent, has completed a phase III trial for the treatment of metastatic colorectal cancer [138]. Interestingly, there are several FDA-approved drugs, such as celecoxib [139] and pyrimethamine [140], that were not initially considered STAT inhibitors. Rather, these drugs were found to inhibit STAT3 and are currently being studied as new treatments for colon and rectal cancers and small lymphocytic lymphoma [141]. JAK 104 is another therapeutic target for inhibition of STAT signaling that has entered clinical application [142]. AG490, a selective inhibitor of the JAK2/STAT3 pathway, was found to inhibit the growth and invasion of gallbladder cancer cells in a recent study [137]. Fedratinib, an FDA-approved JAK2 inhibitor, was used as a co-treatment for P-glycoprotein-overexpressing patients with multidrug-resistant cancer [137]. The findings suggested that targeting the JAK/STAT signaling pathway could be useful for effective cancer therapies and treating multidrug-resistant cancer cells.

### 7.3. p53 Domain Factors

The p53 domain subtype is identified as a beta-sandwich structure with exposed loops and a loop-sheet-helix motif, where one of the loops contacts an arginine residue in the minor groove and the loop-sheet-helix motif interacts with the DNA’s major groove [9,10,11].

#### p53

p53 is a well-known sequence-specific tumor-suppressing transcription factor encoded by the TP53 gene that is required for cell growth and tumor prevention. The TP53 gene is found mutated (mostly missense mutations [143,170]) in various types of cancers [144], and the mutated proteins are unable to bind DNA effectively, causing cells to lose control over cell cycle regulation and apoptosis. TP53 is associated with the activation of EMT in several cancers, such as bladder, prostate, lung, and esophagus. Wild-type p53 (wtp53) is reported to be a key factor for suppressing the EMT process and inhibiting the cancer cells’ metastasis. In contrast, mutant p53 (mutp53) acts as an activator of the EMT, promoting metastasis by affecting EMT-related TFs. Overall, numerous studies indicated that p53 can impact the function of EMT-related proteins such as N-cadherin, Vimentin, Snail, and ZEB1 by regulating the transcription of the genes that encode these factors, interacting with various signaling pathways involved in EMT regulation, and affecting EMT-TF activity on a post-translational level [145,146,147,148].

Several strategies targeting p53 exist for cancer treatment, including preventing the degradation of wtp53, suppressing mutp53, and reactivating the wild-type functions of mutp53 [149]. Aside from the development of small-molecule drugs that free p53 from inhibition by its negative regulators such as MDM2, immunotherapies that aim to improve the human immune system’s ability to recognize and eradicate cancer cells with deregulated p53 have sparked considerable interest in recent years. Vaccination aimed at increasing cellular immunity against p53-contained cancer cells was initiated in the 1990s [171], and the concept of p53 mRNA vaccine has been revived in recent years as a result of the inspiring results of mRNA vaccination in the coronavirus disease 2019 (COVID-19) pandemic. According to the findings of Ma’s team, p53 is an effective antigen for the development of anti-glioma mRNA vaccines [172]. Although none of the drugs targeting p53 have received FDA or EMA approval, constant progress is being made toward better p53-based cancer therapy, and there are numerous clinical trials ongoing currently underway. Clinical trials are currently underway for p53-based gene therapy, p53 immune-based therapy, MDM2– inhibitory small molecules, dual MDM2–MDM4 inhibitory small molecules, mutant p53-targeting small molecules, and restoring p53 structure. Some of the trials are at phase III of development (Milademetan MT (NCT04979442) for liposarcoma, KRT-232 MT (NCT03662126) for myelofibrosis, and APR-246 plus azacytidine (NCT03745716) for myelodysplastic syndrome). Although there are still many challenges to finding efficient and selective p53-targeted drugs that can enter the clinic eventually, p53 gene therapy has great potential for cancer treatment and will hopefully become more effective and widely available in the near future [150].

### 7.4. Runt Domain Factors

The runt domain factors were discovered because they resembled a specific region of the Drosophila protein runt. The runt domain, found in RUNX1, is composed of 12 beta strands, seven of which form an immunoglobulin-like beta-sandwich fold (S-type Ig fold). This fold is also found in the DNA-binding domains of other TFs, including NF-κB, NFAT, p53, STAT, and the T-domain, and is preceded by an alpha-helix at the N-terminus. The runt domain performs two functions: DNA binding and heterodimerization, which occur at different locations within the domain. When RUNX1 and CBFbeta heterodimerize, the runt domain undergoes conformational changes (S-switch) that stabilize DNA binding [9,10,11].

#### RUNX1

Runt-related transcription factor 1 (RUNX1) is a member of the core-binding factor family of TFs that regulates the proper development of many cell lineages. It is involved in several signaling pathways, including the FAK-Src signaling pathway and the p38/MAPK signaling pathway. RUNX1-mediated modulations to hallmarks of cancers consist of cancer stem cell-renewal and self-renewal, cell proliferation, angiogenesis, tumor metastasis, the resistance of chemotherapies, and inhibition of cell apoptosis, resulting in carcinogenesis. RUNX1 is highly expressed in cervical cancer, and its overexpression is associated with a poorer outcome and stronger invasive ability of cancer cells through induction of EMT [152]. By activating the Wnt/β-catenin signaling pathway, RUNX1 enhances the EMT of colorectal cancer cells [153]. In addition, RUNX1 facilitated the TGF-β-induced partial EMT by enhancing the transcription of the PI3K subunit p110δ, which induced renal fibrosis [154].

The overexpression of RUNX1 in human malignancies was significantly associated with the prognosis of patients with cancers such as mesothelioma, lung squamous carcinoma, and stomach adenocarcinoma. It was also positively correlated with infiltrating levels of cancer-associated fibroblasts and modulating chemo-drug resistance in acute myeloid leukemia, colorectal cancer, and ovarian cancer [155,156]. Furthermore, RUNX1 somatic mutations and chromosomal rearrangements are frequently observed in hematological malignancies such as acute myeloid leukemia (AML), ALL, and chronic myelomonocytic leukemia [157]. Interestingly, a recent study by Ariffin et al. indicates that the upregulation of RUNX1 in breast cancer is associated with better outcomes and increases relapse-free survival, providing insight into developing novel therapeutic strategies for BC [156,158]. In AML, by treatment with a BET protein inhibitor or degrader (BET–proteolysis targeting chimera), RUNX1 and its targets are regressed, inducing apoptosis and promoting survival of mice engrafted with AML expressing mutant RUNX1 [159]. Currently, there is no FDA-approved-RUNX1 targeted therapy; however, RUNX1 can act as a promising biomarker for predicting the outcome of various types of cancer, and the targeted therapies of RUNX1 in cancer have immediately piqued the interest of scientists worldwide

## 8. Targeting TFs of Beta-Hairpin Exposed by an Alpha/Beta-Scaffold

### 8.1. SMAD/NF-1 DNA-Binding Domain Factors

The alpha/beta-structured scaffold of the DBDs in this superclass exposes a beta-hairpin, which serves as the primary DNA-contacting element and inserts into the major groove of the DNA.

#### SMAD

Several members of the SMAD family, including R-SMADs, Co-SMADs, and I-SMADs, are further subclassified into SMAD1–9. The R-SMADs and Co-SMADs are composed of Mad homology (MH)1 and MH2 domains located at the amino-terminal and carboxy-terminal ends, respectively, and separated by a flexible linker region. Phosphorylation of R-SMADs by type I receptors takes place on two serine residues at the C-termini that form the SXS sequence motif. SMAD is associated with TGF-β signaling in cancer [160]. SMAD is triggered by the downstream activation of the TGF-β receptor and plays a key role in transducing the TGF-β induced signals from the cytoplasm to the nucleus. Upon activation, SMAD proteins translocate to the nucleus, functioning as TFs and regulating various cellular functions by inducing EMT [161]. The TGF-β/SMAD pathway can be seen in several cancers, such as lung adenocarcinoma [162], esophageal squamous cell cancer [163], and HCC [164].

There are several strategies targeting SMAD to reduce the activation of TGF-β signaling. According to a report, paclitaxel significantly suppresses the TGF-β/SMAD signaling pathway by inhibiting Smad2 phosphorylation in the peritoneum. Moreover, in preclinical development, SB-431542 and SB-505124 (developed by GlaxoSmithKline) are notable TGF-β receptor kinase inhibitors that can hinder Smad2/3 phosphorylation. According to Fenaux et al., Luspatercept, a protein derived from IgG, is linked to a recombinant fusion protein derived from human activin receptor type IIb (ActRIIb) and reduces signaling of SMAD2 and SMAD3 by binding to TGF-β ligands. In patients with lower-risk MDS with ring sideroblasts who regularly received red-cell transfusions and had disease refractory to erythropoiesis-stimulating agents or who discontinued such agents, Luspatercept reduced anemia severity [165,166].

## 9. The Epigenetic Regulation Pathways of TFs Involved in EMT

There are various transcriptional pathways that regulate the EMT, which ultimately leads to the downregulation of E-cadherin and dissolution of cell–cell adhesion. Many genetic or non-genetic regulation pathways were involved in the over-expression of these TFs, which in turn increases the expression level, turnover time, and activities of TFs. Among these pathways, there are certain key pathways that play crucial roles in this process, such as epigenetic regulation, while facilitating chromatin remodeling and transcription initiation through histone H3 acetylation. In this context, we have listed the key epigenetic pathways involving SNAIL, ZEB, Twist, or ISX TFs.

### 9.1. SNAIL-Associated Regulation Pathway

The upregulation of Snai1 is regulated by multiple signaling pathways such as TGFβ, Wnt, and ISX [100,173]. Snail is reported to repress gene expression by binding to the E-cadherin promoter through its carboxy-terminal zinc-finger domains. This binding reduces cell–cell adhesion in cancer cells, facilitating their detachment from the primary tumor and promoting subsequent metastasis. In detail, Snai1 recruits the Polycomb repressive complex 2 (PRC2), which consists of methyltransferases enhancer of zeste homologue 2 (EZH2), G9a, and suppressor of variegation 3–9 homologue 1 (SUV39H1), as well as the co-repressor SIN3A, histone deacetylases 1, 2, and/or 3, and the Lys-specific demethylase 1 (LSD1), upon binding to the E-box sequence in the promoter region. All of these components work together to regulate histone modifications, specifically methylation and acetylation, at specific sites on histone H3, including lysine 4 (H3K4), lysine 9 (H3K9), and lysine 27 (H3K27). The methylation of H3K9 and H3K27 is associated with repressive chromatin, whereas the methylation of H3K4 and acetylation of H3K9 mark active chromatin. This creates a poised state for the promoter, enabling timely activation while maintaining repression in the absence of differentiation signals. The bivalent control of the E-cadherin promoter may contribute to the reversible nature of EMT. Apart from repressing epithelial genes, Snai1 also triggers the activation of genes associated with the mesenchymal phenotype. This mechanism may involve the presence of bivalent domains, which exhibit repressive H3K9 trimethylation and activating H3K18 acetylation. These bivalent domains facilitate the expression of the mesodermal gene goosecoid in response to TGFβ-related Nodal55 [4,83].

### 9.2. Twist-Associated Regulation Pathway

TWIST expression can be activated by several pathways such as Wnt and hypoxia-inducible factor 1α (HIF1α). Under hypoxic conditions, HIF1α can directly bind to TWIST through hypoxia-responsive elements in the TWIST proximal promoter, leading to the upregulation of TWIST expression, and promotes the EMT and the dissemination of tumor cells; additionally, in drosophila melanogaster epithelia, Twist expression is induced by mechanical stress in a β-catenin70-dependent manner.

In cancer cells, Twist1 suppresses E-cadherin and stimulates N-cadherin expression in a SNAIL-independent manner. Twist1 accomplishes this by recruiting the methyltransferase SET8 (also known as SETD8 in humans), which mediates H4K20 monomethylation. This histone modification is associated with repression at E-cadherin promoters and activation at N-cadherin promoters, contributing to the induction of the EMT process [174].

### 9.3. ZEB-Associated Regulation Pathway

ZEBs are also a key regulator in promoting EMT as they repress epithelial cell markers and activate the expression of mesenchymal biomarkers. There are several pathways regulating the ZEBS expression including estrogen signaling cascades, TGFβ, and Wnt/β-catenin signaling pathways. In addition, Twist1 and Snail1 are noted for their cooperative role in regulating the expression levels of ZEB1.

The ZEB-mediated transcriptional pathway involves the recruitment of the C-terminal-binding protein (CTBP) co-repressor, polycomb proteins, CoREST, and the Switch/sucrose non-fermentable (SWI/SNF) chromatin remodeling protein BRG1, which enables ZEB1 to bind to regulatory gene sequences at E-boxes and represses the expression of E-cadherin. Further, ZEB1 expression results in the suppression of various genes associated with the generation and maintenance of epithelial cell polarity. Notable examples of these genes include CDH1, Lgl2, PATJ, and Crumbs3. The expression of ZEB1/2 in epithelial cells induces EMT and promotes a mesenchymal phenotype, thereby facilitating tumor invasion and metastatic dissemination into a cancer stem cell state [4,83,175].

### 9.4. Intestine-Specific Homeobox (ISX) and P300/CBP-Associated Factor (PCAF)

Intestine-specific homeobox (ISX) is a homeobox-containing protein that belongs to the paired subfamily and is homologous to Pax3, Pax7, and Prrx1 phylogenetically [176]. ISX was induced by the pro-inflammatory cytokine interleukin-6 and was highly expressed as a proto-oncoprotein in hepatoma cell and HCC samples [99]. Further, ISX transcriptionally regulated the downstream cell cycle proteins cyclin D1, E2F1, and indoleamine 2, 3-dioxygenases [102,103]. This phenomenon then dysregulated tyrosine catabolism and reduced the levels of immune checkpoint regulators (PD-L1 and B7-2) and epithelial–mesenchymal transition (EMT) regulators (Twist1 and Snail1), thereby affecting the survival time of patients with HCC [102]. Pathologic studies revealed that ISX exhibited a tumor-specific expression pattern, and it is significantly correlated with patient survival and tumor size, number, and stage [99]. Histone modification by acetylation is critical in the regulation of oncogenic gene expression and subsequent cancer progression [177]. Recently, Wang et al. discovered P300/CBP-associated factor (PCAF) acetylation of intestine-specific homeobox (ISX) regulates epithelial–mesenchymal transition (EMT) marker expression and promotes cancer metastasis [100,178]. PCAF acetylation of ISX at lysine residue 69 recruits acetylated bromodomain-containing protein 4 (BRD4) at lysine residue 332, and the resulting complex translocated into the nucleus and binds to EMT promoters, where acetylation of histone 3 at lysine residues 9, 14, and 18 initiates chromatin remodeling and subsequent gene expression in tumor cells [100]. Activated ISX then enhanced EMT marker expression—including TWIST1, Snail 1, and VEGF—and consequent cancer metastasis, but suppressed E-cadherin expression [100,101]. Evidence suggests that the PCAF–ISX–BRD4 regulation axis may hold the promise as a new therapeutic target for the discovery of new small molecular inhibitors, leading ultimately to more efficacious cancer therapy.

## 10. Therapeutic Implications of Targeting EMT-TFs

There are multiple therapeutic strategies for targeting EMT, including the inhibition of upstream signaling pathways such as TGFβ, NF-κB, Wnt, EGFR, and Notch. Additionally, targeting molecular drivers of EMT, such as the key TFs Snai1, ZEB, and Twist, and focusing on mesenchymal cells, integrins, and the extracellular matrix are other approaches. Moreover, there are several therapeutic agents that target TFs for cancer treatment, including small molecule inhibitors, micro RNA, and gene editing techniques [179].

Small molecule inhibitors undergoing clinical trials includes Curcumin (phase III, targeting NF-kB in brain tumor) [180], Metformin [181] (phase III, targeting ZEB1, Slug, Twist in breast cancer), Omo-103 [182,183] (phase I), and Disulfiram [184] (phase II, targeting ERK/NF-kB/Snail pathway in germ cell tumors). Furthermore, BRD4, a member of the bromodomain-contained protein family, is known to be involved in tumorigenesis via its binding to acetylated histones in several types of cancers. Blockade of the BRD4 interaction with HATs by small molecule inhibitors has been shown to effectively block cell proliferation in cancers, some of which have, in fact, been evaluated in human clinical trials. Small molecule inhibitors targeting the bromo- and extra-terminal (BET) domain protein, known as BETi, offer another novel strategy to inhibit the BRD4-MYC axis and subsequently suppress the downstream trans criptional pathway [185]. MicroRNAs have currently been recognized as novel target for EMT-TF; for example, miR-200 [186,187] (Phase II SWOG S0925, targeting ZEBs in prostate cancer), miR-186 [188] (pre-clinical, targeting Twist in cholangiocarcinoma cells), and miR-342 [189] (pre-clinical, targeting FOX in nasopharyngeal carcinoma). CRISPR/Cas gene editing [190] also hold the potential for regulating the EMT-TFs (Phase I, NCT02793856) [191].

## 11. Challenge of Targeting EMT-TFs in Cancer

Though targeting EMT-TFs would be attractive and several agents are undergoing clinical trials, currently there are no FDA (U.S. Food and Drug Administration)-proved therapies for targeting EMT-TFs since there are multiple technical problems remain challenging. First, the expression levels of various EMT-TFs are intricately interconnected through multiple feedback mechanisms. Second, certain EMT-TFs exhibit complementary and redundant functions. The specific role of each EMT-TF greatly relies on the cellular context and microenvironment. Moreover, there is a concern that targeting EMT-TFs with small molecule inhibitors may encounter side effects due to the essential role in normal cell survival and proliferation; further, there remains difficulty in selectively targeting TFs without affecting other TFs since that most of the TFs may interact with each other in multiple pathways. Therefore, the better understanding of the precise regulatory networks and functions of the EMT-TFs in different EMT contexts is needed [192,193].

## 12. Conclusions

The importance of cancer-targeted therapies development is gaining increasing attention in modern society; curing cancer is not just a lofty goal that scientists are attempting to achieve. EMT plays an important role in cancer progression, and numerous TFs regulate the process; the key TFs that mediate the EMT include the Snail-like family, ZEB, and Twists. Currently, targeting TFs that lead to carcinogenesis is a promising novel strategy for cancer therapy; although it is thought to be difficult to target TFs due to their lack of surface involutions and hydrophobic pockets, the invention of compounds, small molecules, or miRNA are used to influence on the multiple functions of TFs, such as protein–protein interaction, tumor microenvironment mediation, cancer cell reprogramming, proliferation, as well as chemo-resistance. Although there are multiple targeted strategies with potential for cancer treatment, including small molecules and miRNA, there are currently no drugs approved by the FDA (U.S. Food and Drug Administration). This is mainly due to the complexity of the EMT pathway, which presents a significant challenge. Therefore, there is no easiest way for an EMT-TF targeted strategy, and further investigation and clinical trials are needed.

Furthermore, considering the complexity and heterogeneity of tumors, as well as the diverse expression of EMT-TFs among individuals, it is clear that the paradigm of precision medicine in cancer therapy requires personalized treatment tailored to each individual patient. This approach is essential for improving clinical outcomes in the future.

In conclusion, this study provides a systematic classification of various types of TFs that involved in the EMT process based on their DBD structure and highlights some of the main TFs of the superclass and subclass that have the potential to be cancer biomarkers or targeted therapies. There are several superclasses (including beta-barrel DNA-binding domains, targeting TFs of beta-sheet binding to DNA, and yet undefined DNA-binding domains) and subclasses have not been discussed or mentioned in this paper due to the scarce relevant studies, but some of these TFs still hold great potential for cancer-targeted therapies (for example, non-POU domain containing octamer binding domain factor), and further evaluation and clinical trials are required.

## Figures and Tables

**Table 1 cancers-15-03338-t001:** The character of superfamily and subfamily class of transcription factors and their EMT represented gene. The structure of TF DNA-binding domains is cited from the study of Edgar Wingender et al. [9,10,11].

Domain	Domain Character	Class	Class Character	Representation	References
Basic Domain 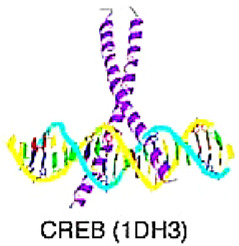	TFs that are part of this superfamily interact with DNA by means of a basic region, which is disordered in solution and folds into an alpha-helix when binding to DNA.	basic motif leucine zipper (bZIP) factors	Characterized by a leucine zipper region that mediates dimerization with another bZIP domain and a basic region that binds to DNA.	JUN, FOS, BACH1	[9,10,11]
		bHLH factors	DNA-binding basic region followed by a motif of two potential amphipathic alpha-helices connected by a loop, possibly an omega loop.	E2A, TWIST	
Zinc-coordination DNA-binding domain 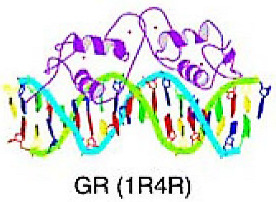	The zinc-coordinating domains are characterized by the presence of a zinc ion that is coordinated by two or more conserved cysteine or histidine residues in the protein structure.	Nuclear receptors with C4 zinc fingers	C4 zinc finger motif consists of four cysteine residues coordinating one zinc ion and binds to DNA through the recognition of specific DNA sequences known as hormone response elements. In each molecule of the nuclear receptor, there are two DNA-binding motifs that are different in size, composition, and function. The first zinc finger binds to DNA through the major groove, and the second zinc finger mediates dimerization upon DNA binding, with an alpha-helix conformation.	AR, RAR	
		C2H2 Zinc finger factors	Feature of zinc finger motif of TFIIIA/Krueppel type, consisting of two cysteine and two histidine residues coordinating a zinc ion, with some cases replacing a histidine with another cysteine. This zinc ion is crucial for DNA binding. Typically, the first half of the finger sequence is arranged in two antiparallel beta-strands, while the second half is organized as an alpha-helix and partially as a 310-helix. The conserved phenylalanine and leucine residues create hydrophobic contacts between the beta-strands and the alpha-helix, which binds to DNA via the major groove.	Snail-like	
Helix-turn-helix domain 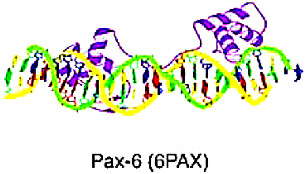	The helix-turn-helix domain composed of two alpha helices connected by a short beta turn, forming a “V” shape. The first helix is often referred to as the recognition helix, as it makes specific contacts with the DNA. The second helix helps stabilize the structure of the domain and interacts with other proteins in some cases.	Homeo Domain factors	Made up of a series of three alpha-helices in a row, where the third helix predominantly interacts with the major groove of the DNA, and some interactions with the minor groove can also be seen. The homeodomain binds to DNA as a monomer, recognizing short DNA sequences typically 5–8 base pairs in length, and often functions in transcriptional regulation.	ZEB, ISX	
		Fork head/winged helix factors	The DBD is about 110 amino acids long. It has three closely packed alpha-helices, where the third alpha-helix is exposed towards the major groove of the DNA. The domain also makes minor groove contacts. When it binds to DNA, it causes a bend of 13 degrees.	FOX	
Alpha-helical DNA-binding domains 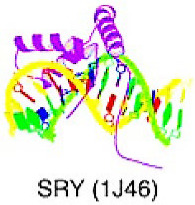	The superclass includes DBDs that exhibits alpha-helically structured interfaces which interacting with the DNA.	HMG domain factors	The proteins with a HMG domain shared an identical structure, the HMG box. This domain shows a typical L-shaped conformation composed of three alpha-helices and an extended N-terminal extension of the first helix.	SOX	
Beta-core (Immunoglobulin fold) 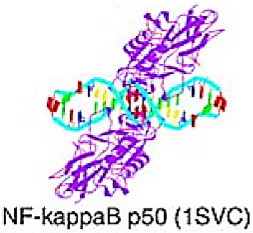	The DNA-binding domains in this superclass possess an immunoglobulin-like structure, consisting of a beta-core with a beta-sandwich architecture. The DNA-contact interface is primarily composed of loops, but may also include other elements of secondary structure, with DNA-binding residues extending from this interface.	Rel homology region factors	The structure of the Rel-type protein shows a bipartite subdomain structure, with each subdomain comprises a beta-barrel with five loops that form an extensive contact surface to the DNA’s major groove.	NF-κB	
		STAT domain factors	The DNA binding motif of STAT proteins involves a dimeric organization with an eight-stranded beta-barrel and a four-helix bundle at the N-terminus, followed by an alpha-helical connector region at the C-terminus.	STAT	
		p53 domain factors	The p53 domain subtype is identified as a beta sandwich composed of a scaffold in addition to several loops and a loop-sheet-helix motif. One of the loops forms a contact of an arginine residue in the minor groove of the DNA, and side chains of the loop-sheet-helix motif in the major one.	P53	
		Runt domain factors	The Runt domain is composed of 12 beta strands, with seven forming an immunoglobulin-like beta sandwich fold (S-type Ig fold), and is preceded by an alpha helix at the N-terminus.	RUNX1	
Beta-sheet binding to DNA 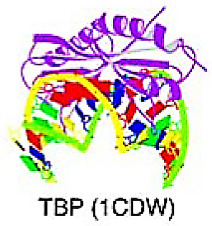	The DBD of this superclass attach to DNA using either individual elongated strands or beta-sheets	TATA-binding proteins (TBP)	The structure of TBP involves a 10-stranded beta-sheet that forms a symmetrical saddle shape, with four alpha-helices on the convex side and hydrophobic interactions on the concave side, which bind to the minor groove of the TATA-box. This interaction causes a noticeable bend in the DNA helix.	TBP	
Beta-hairpin exposed by an alpha/beta-scaffold 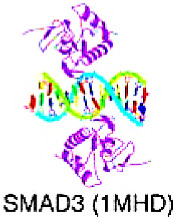	The alpha/beta-structured scaffold of the DNA-binding domains in this superclass reveals a beta-hairpin, which acts as the primary DNA-binding element by inserting into the major groove of the DNA.	SMAD/NF-1 DNA-binding domain factors	The alpha/beta-structured scaffold of the DBDs in this superclass exposes a beta-hairpin, which serves as the primary DNA-contacting element and inserts into the major groove of the DNA.	SMAD	

**Table 2 cancers-15-03338-t002:** The Consensus binding sequence of represent TF and their oncogenic effects and current findings.

Domain	Class	Representation	Consensus Binding Sequence	Oncogenic Effects and Current Findings	References
Basic domain	bZIP	JUN	TGAGTC	Subunit of AP-1, involved in EMT mechanism in lung adenocarcinoma, nasopharyngeal carcinoma, prostate cancer, and breast cancer. Potential of cancer therapeutic targets: ■ Antagonizing DNA TRE sites (MLN944, SR11302and Veratramine). ■ Antagonizing the c-Jun DBD (T-5224). ■ Antagonizing the Lucine zipper interface. ■ Antagonizing the full bZIP domain.	[9,19,29,30,31,32,33,34]
		FOS	TGAGTC		
		BACH1	GCTGAG	Regulating the homeostasis of iron and related reactions, regulating oxidative stress response, enhancing cancer cell metastasis, promoting aerobic glycolysis, and promoting EMT. ■ BACH1 regulates additional TF genes that are involved in the EMT process such as FOXA1, CDH2, and SNAI2.	[2,35,36,37,38,39,40,41]
	bHLH	E2A	CAGGTG	Involved in the EMT and is associated with the formation of breast cancer, colorectal cancer, ovarian cancer, cervical carcinoma, and acute lymphoblastic leukemia. E2A-PBX1 fusion protein and become a coactivator for RUNX1 in the t(1,19) chromosomal translocation of patients with acute lymphoblastic leukemia, resulting in the unfavorable B-cell development.	[9,42,43,44,45,46,47,49]
		TWIST *	CGTCTG	TWIST is one of the major regulators of EMT. Twist-1 and Twist-2 can act as molecular switches to activate or suppress target genes through direct or indirect mechanisms. Both Twist-1 and Twist-2 induce EMT and regulate the expression of EMT-associated genes or downstream effector, such as N-cadherin or Epithelial membrane protein 3 (EMP3). The overexpression of Twist-1 and Twist-2 involved in cancer cell migration, stem cells’ self-renewal, multiple drug resistance, cell apoptosis, and immune surveillance.	[50,51,52,53]
Zinc-coordination DNA-binding domain	Nuclear receptors with C4 zinc fingers	AR	AGAACA	AR is involved in the EMT process of prostate cancer, breast cancer, as well as bladder cancer, and is associated with the tumor metastasis and advanced cancer stages. Various novel therapeutic agents, such as the anti-diabetic drug metformin or the selective estrogen receptor modulator ormeloxifene, have been tested in pre-clinical studies for their potential to inhibit EMT in prostate cancer.	[65,66,67,69,70,71]
		RAR	TGACCT	The interruption of RA signaling pathways is believed to be responsible for the development of various hematological and non-hematological cancers. The role of RARα in EMT is controversial: ■ RARα downregulate the EMT processes in retinal pigment epithelial cells and renal tubular cells. ■ The overexpression of RARα increases the mRNA levels of well-known EMT-inducing factors such as SLUG, FOXC2, ZEB1, and ZEB2 in mammary epithelial cells.	[74,75,76,79]
	C2H2 Zinc finger factors	Snail-like *	CACCTGA	The Snail-like TF subfamily is the key TFs that mediated the EMT, and it is consisted of several members including Snai1 (Snail), Snai2 (Slug) and Snai3 (Smuc). One of the most well-known family that plays a crucial role in EMT, regulating various cellular processes such as cellular differentiation, cell movements, and overall survival in cancers Targeted strategies: ■ CYD19: targets Snail and successfully disrupted CREB-binding protein (CBP)/p300-mediated Snail acetylation by binding to Snail. ■ Thiolutin: suppress PSMD14/SNAIL axis, thus decreasing the EMT process. ■ CAMKK2 or ACLY: interrupt the autophagy/acetyl-CoA/acetyl-Snail axis, leading to the inhibition of lung cancer metastasis.	[81,82,83,84,85,89,90,91]
Helix-turn-helix domain	Homeo Domain factors	ZEB *	CACCTG	Key TFs that mediate the EMT process and consist of ZEB1 and ZEB2. The activation of MEK1/2, ERK1/2, Fos-related antigen 1 (Fra-1), and TGF-β enhance the expression of both ZEB1 and ZEB2, increasing the tumor invasion. Targeted strategies: ■ microRNAs: the miR/ZEB1 axis can be regulated by lncRNAs or circRNAs, which therefore regulate tumor malignancy in several cancers. ■ NR4A1: reduce the expression of ZEBs, which inhibits the TGF-β-Smad2/3/4-ZEB signaling pathway, thus inhibiting the EMT-induced liver fibrosis.	[92,93,94,95,96,97]
		ISX	CTAATT	Plays an important role in hepatocellular carcinoma(HCC) ■ Organize the feed-forward mechanism of immune suppression that involves kynurenine-AHR signaling and PD-L1 and offer the function of immune escape. ■ Activated E2F transcription factor 1 (E2F1) and upregulate the oncogenic activity. ■ Involved in ISX-PCAF-BRD4 complex that mediates EMT signaling and regulates tumor initiation and metastasis by regulating Twist1 and SNAI1.	[9,98,99,100,101,102,103]
	Fork head/winged helix factors	FOX	TGTTT(A/G)	FOXs in EMT: ■ FOXG1: enhance the EMT process in HCC cells by inducing the nuclear transport of β-catenin and subsequently retaining it in the nucleus. ■ FOXC1, FOXC2, FOXK1, FOXQ1, and FOXM1: associated with tumor metastasis and poorer outcome through regulating TGFβ-induced EMT process in various types of cancers. Strategies of regulating FOX expression: ■ microRNA (miR-342, miR-204 and miR-1269) ■ RNA interference ■ Proteasome Inhibitors ■ Genistein ■ Peptide inhibitors or thiazole antibiotics	[4,9,104,105,106,107,108,109,110]
alpha-helical DNA-binding domains	HMG domain factors	SOX	AACAAT	The dysregulation of SOXs is observed in almost every type of human cancers, including breast cancer, prostate cancer, liver cancer, renal cell carcinoma, thyroid cancer, cervical cancer, brain tumor, gastrointestinal, and lung cancer. The role of SOXs in EMT ■ SOX4: Regulate target genes associated with mesenchymal features, including N-cadherin, ADAM10, TMEM2, TNC, FZD5, neuropilin-1, and semaphorin-3A, to promote EMT. ■ SOX2: Regulate EMT during the development of cranial neural crest cells (CNCCs) and impacts the fate of cells involved in head growth during neural crest development. ■ SOX9, 10, and 11: associated with increased tumor metastasis and worse overall survival by regulating the ability of cancer cells to undergo EMT and acquire mesenchymal characteristics. SOX2 is the most promising target currently in the SOX family ■ Clinical trials targeting SOX2 ♦ Phase I/II: SAHA ♦ Phase I SOX2-derived peptide	[9,111,112,113,114,115,116,117,118,119,120,121,122]
Beta-core (Immunoglobulin fold)	Rel homology region factors	NF-κB	NF-κB p50-like: GGGAATNF-κB p65-like: GAAAAT	NF-κB plays a vital role in infection response and cell survival, differentiation as well as controlling regulators of apoptosis, stress-response genes, cytokines, chemokines, growth factors, and their receptors. NF-κB pathway is involved in the induction of EMT in glioblastoma, BC, and nasopharyngeal carcinoma, contributing to tumor progression as well as treatment resistance. Drug targeting the NF-κB for cancer: ■ FDA approved: Bortezomib ■ Under clinical trial: Acalabrutinib, Ibrutinib, Dasatinib, and LCL-161	[9,123,124,125,126,127,128,129,130]
	STAT domain factors	STAT	TTC(N2-4)GAA	STAT signaling pathway is involved in cell differentiation, proliferation, and inflammation and is found to be widely implicated in various types of cancer. Activation of IL-6/JAK2/STAT3 pathway promotes metastasis by increasing the expression of EMT-inducing TFs such as ZEB1, Snail, JUNB, and Twist-1. Drug targeting the STAT for the cancer: ■ Direct inhibitors of STAT: OPB-31121, OPB-111077, OPB-51602, Napabucasin (BBI-608), Celecoxib, Pyrimethamine ■ Inhibitor of the JAK2: Fedratinib (FDA approved)	[9,131,132,133,134,135,136,137,138,139,140,141,142]
	p53 domain factors	p53	RRRCWWGYYY-NNN-RRRCWWGYYY(R = A or G, W = A or T, Y = C or T, and N = any nucleotide.)	p53 is a well-known sequence-specific tumor suppressing transcription factor encoded by TP53 gene and is critical for cell growth as well as tumor prevention. TP53 is associated with the activation of EMT in several cancers, and impacts the function of EMT-related protein, interacting with various signaling pathways involved in EMT regulation, and affects EMT-TF activity on a post-translational level. Presently, active clinical trials include p53-based gene therapy, p53 immune-based therapy, MDM2– inhibitory small molecules, dual MDM2–MDM4 inhibitory small molecules, mutant p53-targeting small molecules and restoring p53 structure; however, none of the p53 drugs have so far made it to FDA or EMA approval.	[9,143,144,145,146,147,148,149,150,151]
	Runt domain factors	RUNX1	TGTGGTTAAC	Runt-related transcription factor 1 (RUNX1) is a member of the core-binding factor family of TFs, regulating the proper development in many cell lineages. RUNX1 is noted to enhance EMT through activating the Wnt/β-catenin signaling pathway, and facilitating the TGF-β-induced partial EMT by enhancing the transcription of the PI3K subunit p110δ, which induces renal fibrosis Somatic mutations and chromosomal rearrangements involving RUNX1 are frequently observed in hematological malignancies such as acute AML, ALL, and chronic myelomonocytic leukemia. The upregulation of RUNX1 in breast cancer is associated with better outcome and increases the relapse free survival.	[9,152,153,154,155,156,157,158,159]
beta-hairpin exposed by an alpha/beta –scaffold	SMAD/NF-1 DNA-binding domain factors	SMAD	GTCTAGAC	There are several members of the SMAD family, including R-SMADs, Co-SMADs, and I-SMADs, which are further subclassified into SMAD1 to SMAD9. SMAD is triggered by the downstream activation of TGF-β receptor, regulating various cellular functions by inducing EMT. Targeted strategies: ■ Paclitaxel: inhibits Smad2 phosphorylation in the peritoneum, thus suppress the TGF-β/SMAD signaling pathway ■ SB-431542 and SB-505124: TGF-β receptor kinase inhibitors that can hinder Smad2/3 phosphorylation ■ Luspatercept: reduces signaling of SMAD2 and SMAD3 by binding to TGF-β ligands	[160,161,162,163,164,165,166]

* Represents as master regulating factors in EMT process.

## Data Availability

All data supporting the findings of this study are available within the article and from the corresponding author upon reasonable request.

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
