# Peer review of "Recent Advances in Transcription Factors Biomarkers and Targeted Therapies Focusing on Epithelial–Mesenchymal Transition"

_cancers, 2023, doi:10.3390/cancers15133338_

Round 1

Reviewer 1 Report (Previous Reviewer 2)

I must admit that the change of the concept of the article by the authors is an incredibly clever play, which turned out to be good for the article itself. Nevertheless, the authors need to delve even further into the literature.

1.       The authors should in more detail describe the EMT in cancers, and how this process is associated with progression and prognosis. Furthermore, the authors should mention that though huge amount of data showing the importance of EMT in cancers, its role in vivo may not be all that important. Please read and cite the following articles: doi: 10.1158/0008-5472.CAN-05-0699; doi: 10.1038/nrc.2017.118; doi: 10.1002/emmm.200900043.

2.       Targeting TFs may be on different levels: expression, stability of mRNA, stability of protein, activity by ligands (if TF has a ligand binding domain), or post-translational modification. It would greatly increase the value of this article if the authors check if any of the known and approved anticancer drugs act via these mechanisms on the considered TFs. They can prepare a table describing this topic. Please include clinical trial numbers targeting these TFs if available.

3.       Please describe the role of estrogen receptor (ER), progesterone receptor (PR), and glucocorticoid receptor (GR) in EMT and how these factors can be targeted.

4.       Regarding SNAIs. Many authors indicate that histone deacetylase inhibitors induce the expression of SNAIs and this sensitizes cancers to many drugs. Please explain this phenomenon, read and cite the following articles: doi: 10.1016/j.ejphar.2023.175728; doi: 10.1371/journal.pone.0210889; doi: 10.1038/cddiscovery.2016.41

5.       In the conclusion section, the authors should state what is the perspective for targeting these transcription factors, which group is easiest as a therapeutic target, and in general if it is an easy or difficult target in anticancer therapies.

Author Response

I must admit that the change of the concept of the article by the authors is an incredibly clever play, which turned out to be good for the article itself. Nevertheless, the authors need to delve even further into the literature.
1. The authors should in more detail describe the EMT in cancers, and how this
process is associated with progression and prognosis. Furthermore, the authors
should mention that though huge amount of data showing the importance of EMT in cancers, its role in vivo may not be all that important. Please read and cite the following articles: doi: 10.1158/0008-5472.CAN-05-0699; doi:
10.1038/nrc.2017.118; doi: 10.1002/emmm.200900043.
Response: We thank the editor’s suggestion. Accordingly, in the new revised
manuscript, it is revised the statement of mechanism and importance of EMT in cancer prognosis and therapy on 1 and 2 of introduction section (blue area) referencing above paper.
2. Targeting TFs may be on different levels: expression, stability of mRNA, stability of protein, activity by ligands (if TF has a ligand binding domain), or posttranslational modification. It would greatly increase the value of this article if the authors check if any of the known and approved anticancer drugs act via
these mechanisms on the considered TFs. They can prepare a table describing
this topic. Please include clinical trial numbers targeting these TFs if available.
Response: We thank the editor’s suggestion. Accordingly, we have added relevant content about the anticancer drugs targeting some studied transcription factors (TFs) at different levels in cancer therapies in the section 10 and part of conclusion (section 12). Actually, the drugs (or approaches) are still rare, even no FDA proved case. Sure, there are some small molecular anti-TFs drugs for cancer therapy on different stages of clinical try, but most of them are still risky. So, we sated this part by statement instead of table.
3. Please describe the role of estrogen receptor (ER), progesterone receptor (PR),
and glucocorticoid receptor (GR) in EMT and how these factors can be targeted.
Response: We thank the editor’s suggestion. Accordingly, we have added relevant discuss content involved EMT regulation and their therapeutic potentials of estrogen receptor (ER), progesterone receptor (PR), and glucocorticoid receptor (GR) in the section 4.1.1 (Steroid hormone receptor (SHR)) in the revised manuscript.
4. Regarding SNAIs. Many authors indicate that histone deacetylase inhibitors
induce the expression of SNAIs and this sensitizes cancers to many drugs. Please
explain this phenomenon, read and cite the following articles: doi:
10.1016/j.ejphar.2023.175728; doi: 10.1371/journal.pone.0210889; doi:
10.1038/cddiscovery.2016.41
Response: We thank the editor’s suggestion. Accordingly, in the new revised
manuscript, it is revised the statement of potential mechanism about how histone deacetylase inhibitors induce the expression of SNAIs, sensitizing cancers to many drugs on 4.2.1 and 9.2 section referencing above paper, but it still one possibility to explain the phenomena.
5. In the conclusion section, the authors should state what is the perspective for
targeting these transcription factors, which group is easiest as a therapeutic
target, and in general if it is an easy or difficult target in anticancer therapies.
Response: We thank the editor’s suggestion. Accordingly, we have added relevant discuss content involved therapeutic potentials of different targets as an overall assessment in Section 10, 11 and conclusion (section 12) in the revised manuscript.

Thank you for your consideration.
Sincerely yours,
Shih-Hsien Hsu, Ph.D.
Professor/Director of Master’s program
Graduate Institute of Medicine
Kaohsiung Medical University,
Kaohsiung, Taiwan
E-mail: jackhsu@kmu.edu.tw

Round 2

Reviewer 1 Report (Previous Reviewer 2)

The authors significantly improved their manuscript and I have no further comments.

This manuscript is a resubmission of an earlier submission. The following is a list of the peer review reports and author responses from that submission.

Round 1

Reviewer 1 Report

In the Review Article entitled “Recent advances in carcinogenesis transcription factors biomarkers and therapies”, Kai-Ting Chuang and Co-authors have reporting a critical analysis of currently available data on the role of transcription factors (TF) in carcinogenesis and targeted therapies. In fact, in this context, the authors have highlighting the potential therapeutic implications of these findings. The topic addressed is very interesting and well discussed by the authors and could give useful information to define new TF-based therapeutic promising approaches.

Collectively, the paper results well written and structured in a clear form and only minor revision is required in order to organize the text accordingly to Instructions for Authors.

Reviewer 2 Report

Chuang et al. in their review try to analyze the role and potential targeting of different transcription factors in cancers. The authors undertook a very difficult and breakneck task. Unfortunately, the breadth of the subject indicates that either the authors will write a book or fail. Unfortunately, in this case, the second scenario came true. The subject matter was presented superficially and without proper study of the literature. The authors omitted a multitude of very important transcription factors in the context of cancer, including. SNAI, AR, RAR. This is completely incomprehensible to me. I advise authors to focus on only one group of transcription factors and describe their role in the context of many cancers or describe all groups of transcription factors in the context of a selected cancer. In this case, the work does not bring anything new.